# Preliminary Report: Rapid Intraoperative Detection of Residual Glioma Cell in Resection Cavity Walls Using a Compact Fluorescence Microscope

**DOI:** 10.3390/jcm10225375

**Published:** 2021-11-18

**Authors:** Jiro Akimoto, Shinjiro Fukami, Megumi Ichikawa, Kenta Nagai, Michihiro Kohno

**Affiliations:** 1Department of Neurosurgery, Kohsei Chuo General Hospital, Tokyo 153-0062, Japan; 2Department of Neurosurgery, Tokyo Medical University, Tokyo 160-8402, Japan; sinjifk@gmail.com (S.F.); norimakiqoo@gmail.com (M.I.); gonguripon@yahoo.co.jp (K.N.); mkouno@tokyo-med.ac.jp (M.K.)

**Keywords:** intraoperative photodiagnosis, malignant glioma, fluorescence–guided surgery, intraoperative cytology, fluorescence microscope

## Abstract

Objective: The surgical eradication of malignant glioma cells is theoretically impossible. Therefore, reducing the number of remaining tumor cells around the brain–tumor interface (BTI) is crucial for achieving satisfactory clinical results. The usefulness of fluorescence–guided resection for the treatment of malignant glioma was recently reported, but the detection of infiltrating tumor cells in the BTI using a surgical microscope is not realistic. Therefore, we have developed an intraoperative rapid fluorescence cytology system, and exploratorily evaluated its clinical feasibility for the management of malignant glioma. Materials and methods: A total of 25 selected patients with malignant glioma (newly diagnosed: 17; recurrent: 8) underwent surgical resection under photodiagnosis using photosensitizer Talaporfin sodium and a semiconductor laser. Intraoperatively, a crush smear preparation was made from a tiny amount of tumor tissue, and the fluorescence emitted upon 620/660 nm excitation was evaluated rapidly using a compact fluorescence microscope in the operating theater. Results: Fluorescence intensities of tumor tissues measured using a surgical microscope correlated with the tumor cell densities of tissues evaluated by measuring the red fluorescence emitted from the cytoplasm of tumor cells using a fluorescence microscope. A “weak fluorescence” indicated a reduction in the tumor cell density, whereas “no fluorescence” did not indicate the complete eradication of the tumor cells, but indicated that few tumor cells were emitting fluorescence. Conclusion: The rapid intraoperative detection of fluorescence from glioma cells using a compact fluorescence microscope was probably useful to evaluate the presence of tumor cells in the resection cavity walls, and could provide surgical implications for the more complete resection of malignant gliomas.

## 1. Introduction:

In the management of the malignant gliomas, the extent of the surgical resection is the most important prognostic factor [1,2,3,4,5]. Therefore, methods to improve the surgical resection rate of these tumors is an important issue for neurosurgeons. Neurosurgeons perform detailed assessments of various information, such as a patient’s clinical profile, the preoperative radiological images, the intraoperative physiological brain functions, the functional brain mapping by awake craniotomy, the optical navigation and intraoperative MRI, to resolve this important clinical issue [1,2,3,4,5,6].

Recently, the surgical usefulness of the fluorescence–guided resection (FGR) method, using a photosensitizer and 5-aminolevulinic acid (5-ALA), has been accepted worldwide, and neurosurgeons confront the malignant gliomas with this novel intraoperative method to improve the extent of resection of these tumors [7,8,9]. However, with the spread of the FGR method, its problems and limitations have become clear, such as the problem of specificity of 5-ALA accumulation in tumor cells, and the problem of the subjective assessment of the intensity of fluorescence by the surgeon’s naked eye under the surgical microscope [9,10]. To overcome these problems, several authors performed basic and clinical studies on the quantification of the 5-ALA metabolite protoporphyrin IX (PPIX) in the brain and tumor tissues using optical spectroscopy, and an intraoperative augmentation method of these optical information using artificial intelligence [11,12,13,14,15].

We investigated the possibilities of performing the FGR method using another photosensitizer, talaporfin sodium (TPS), for malignant gliomas, and reported the selective accumulation of TPS in glioma cells in both in vitro and in vivo studies, as well as having developed an intraoperative photodiagnosis method using TPS with semiconductor laser systems [16,17,18,19,20]. Furthermore, we reported that additional intraoperative photodynamic therapy (PDT) applied to the wall of the resection cavity of malignant gliomas using TPS with a semiconductor laser improves the median of progression-free survival and overall survival of patients, and PDT was subsequently approved for use under insurance by the Japanese Government in 2013 [21,22,23].

Basic research data demonstrated that the intensity of fluorescence emitted from the tumor tissues of a rat xenograft model when TPS was administered was 5- to 8-fold greater than that from a tumor model when 5-ALA was administered [24], and we reported the detection of strong fluorescence intensities emitted from the tumor tissues in our clinical surgeries of malignant gliomas [19]. However, we encountered problems in the subjective assessment of fluorescence emission intensities of tumors and brain–tumor interfaces (BTI) by the naked eye under a surgical microscope, similarly to the evaluation of PPIX [19,20].

To overcome this problem, we developed an intraoperative rapid fluorescence cytology system for the objective assessment of fluorescence emission from the tumor cells, which take up TPS, in the operating theater. In this article, we introduce the detailed method of this system, and report its possibility of clinical feasibility for the improvement of the extent of resection of malignant gliomas, with the demonstration of clinical cases.

## 2. Materials and Methods

### 2.1. TPS and Semiconductor Laser

The photosensitizer TPS (Laserphyrin^®^, Meiji Seika Pharma Co., Ltd., Tokyo, Japan) is a hydrophilic compound synthesized by coupling aspartic acid and chlorine, and is utilized in PDT for primary malignant brain tumors, in combination with a semiconductor laser (PD Laser BT^®^, Meiji Seika Pharma Co., Ltd., Tokyo, Japan) at a wavelength of 664 nm. TPS is a second-generation photosensitizer that is more quickly excreted from the body than the first-generation photosensitizers, and is characterized by the rapid resolution of the skin photosensitivity reaction.

### 2.2. Intraoperative Photodiagnosis Using a Surgical Microscope

TPS was administered as a bolus intravenously at a dose of 40 mg/m^2^ in light-shielded conditions, 24 h prior to surgery. Craniotomy was performed under illumination at ≤500 lux or less. First, the brain surface was observed under halogen light illumination. Then, the halogen light was turned off, and the brain surface was irradiated with a diode laser at 664 nm at a power density of 10 mW/m^2^, with a beam diameter of 40 mm and an irradiation area of 12.6 cm^2^, to observe the presence or absence of tumor fluorescence. The wavelength of fluorescence emitted from tumor tissues was 672 nm, and the fluorescence intensity displayed on the monitor was classified into three grades: strong fluorescence (S); weak fluorescence (W); and no fluorescence (N). Tumor resection was performed by fluorescence image guidance using an optical navigation system (Kolibri^®^, Brain Lab KK, Tokyo, Japan) under physiological monitoring.

### 2.3. Compact Fluorescence Microscope

We used a compact fluorescence microscope BZ-8000 (Biozero^®^, height: 41 cm; width: 31.2 cm; depth: 55.7 cm; weight: 28 kg) developed by Keyence Co., Ltd. (Osaka, Japan) in this study. The microscope was brought into the operating theater and put it in front of the surgeon (Figure 1B). This microscope is a handstand-type phase-contrast fluorescence microscope containing a dark room box, and has high quality performance with full electric control. The following three filter systems were used to observe fluorescence emission from the tumor tissues: PDD-B (excitation: 450/40 nm; absorption: 460 nm; dichromic mirror: 435 nm) was used to observe autofluorescence from the normal brain; PDD-A (excitation: 550/40 nm; absorption: 610 nm; dichromic mirror: 595 nm) for PPIX fluorescence and autofluorescence from 5-ALA metabolites; and Cy5 (excitation: 620/60 nm; absorption: 700/75 nm; dichromic mirror: 660 nm) for TPS. (Figure 1A) The images obtained using each filter system were displayed on the PC monitor using BZ Viewer^®^ (Keyence Co., Ltd., Osaka, Japan), and the surgeons evaluated the presence or absence of fluorescence-emitting tumor cells fluorescence with their own eyes (Figure 1C).

### 2.4. Real-Time, Intraoperative Fluorescence Cytology

#### 2.4.1. Tissue Preparation

Crush smear tissue preparations were made using two nonfluorescent glass slides (Matsunami Glass Ind., Ltd., Osaka, Japan) for a 1 mm piece of tumor obtained under photodiagnosis. Immediately, tissues were fixed using aerosol tissue fixative (Cytokeep II^®^, Alfresa Pharma Co., Ltd., Osaka, Japan) and covered with a nonfluorescent cover glass (Matsunami Glass Ind., Ltd.).

#### 2.4.2. Tissue Assessment by BZ-8000

The room of BZ-8000, and images depicted on the monitor of a personal computer (PC, Windows XP) were observed. Movement of the stage, reduction of the scattering light, focusing, zooming (×10–×400), taking pictures and the changing of the selected filter systems were controlled by the PC. The time required to image observation from tissue sampling was less than 2 min. After the observation of the fluorescence image, hematoxylin and eosin staining of the fixed tissue was performed and histopathological features of the tissue were observed under white light.

#### 2.4.3. Analysis of Fluorescence Images

The obtained fluorescence images on BZ Viewer^®^ were analyzed by the original software BZ-II analyzer^®^, developed by Keyence Co, Ltd., Osaka, Japan. In this study, we counted the number of cells emitting red fluorescence under the Cy5 filter in ×100 images. 

### 2.5. Phantom Experiments

Experiments were performed to analyze whether the newly developed fluorescence microscopy system enabled adequate intraoperative observation of the fluorescence emitted from glioma cells during glioma surgery. TPS at a concentration of 10 μg/mL was mixed with 10% bovine serum albumin (BSA; Wako Pure Chemical Industries Ltd., Osaka, Japan), and then a cotton fiber was moistened with the solution. This cotton fiber was mounted on the nonfluorescent glass slide and fixed by Cytokeep II^®^. The fixed cotton fiber was covered with a nonfluorescent micro glass cover and then the fluorescence emission was observed using Biozero^®^. The other experiment concerned the observation of the autofluorescence of PPIX from iodine egg shells. A small piece (1 mm × 1 mm) of iodine egg shell was mounted onto a glass slide and fixed using Cytokeep II^®^, and then the autofluorescence emitted from the iodine egg shell was observed.

### 2.6. Clinical Cases 

The study subjects were 25 selected patients who received the protocol-specified surgery after being diagnosed with glioma by preoperative diagnostic imaging performed by a single surgeon (JA) between May 2006 to May 2010. Of the 25 patients with glioma, 17 patients had a newly diagnosed tumor, and 8 patients had a recurrent tumor. The histological malignancy grade was grade II in 1 patient, grade III in 7 patients and grade IV in 17 patients. The institutional review board of Tokyo Medical University approved the study (study approval number: No.419), and all patients gave their informed consent before their participation.

### 2.7. Calculation of the Extent of Resection

The extent of resection was determined based on MRI images obtained before surgical resection and within 3 days after the resection. For gadolinium-enhanced tumors, gadolinium-enhanced T1-weighted axial imaging was used. The sum of the products of the perpendicular diameters (SPD) of the contrast-enhanced lesions was calculated. Then, the SPD of the residual lesions on immediate postoperative imaging was determined, and the extent of resection was calculated. For non-gadolinium-enhanced tumors, the SPD of the areas of prolonged T2 on T2-weighted imaging was assessed to calculate the extent of resection.

### 2.8. Statistical Analysis

The TPS-positive tumor cell counts in each tumor tissue calculated from the intensity of fluorescence under photodiagnosis were compared by the Student *t*-test, using Mac statistical analysis software ver. 3 (Misumi Co., Ltd., Tokyo, Japan). A *p*-value of less than 0.05 was considered to indicate a statistically significant difference between two groups.

## 3. Results

### 3.1. Phantom Experiments

After the observation of the structure of the cotton fibers under white light, the green autofluorescence from cotton fibers was observed using the PDD-B filter. From the cotton fibers soaked in a mixture of the TPS solutions and 10% BSA, strong red fluorescence was observed under the PDD-A filter, and the broadband fluorescence could be observed using merged images from the PDD-A and the PDD-B filters. A weak red fluorescence was visible using the Cy 5 filter, which was considered to be the fluorescence emitted from TPS, and not autofluorescence from the cotton fiber (Figure 2A–C).

The green autofluorescence of the membrane of the iodine egg shell was visible using the PDD-B filter, and a strong red autofluorescence was observed from the membrane and shell, suspected to be PPIX, using PDD-A filter. Red fluorescence from the PPIX of the shell was also observed using the Cy 5 filter (Figure 2D–F).

These findings suggested that observing the fluorescence of target molecules within tumor cells using this compact fluorescence microscope together with the appropriate filters was possible.

### 3.2. Intraoperative Detection of Infiltrating Tumor Cells Using Fluorescence Microscopy

Crush smear tissue samples of the tumor bulk, marginal area of the tumor, and adjacent edematous brain tissue were analyzed using a fluorescence microscope coupled with a Cy 5 filter and H & E staining. The tumor bulk emitted a strong fluorescence from the many tumor cells and a weak fluorescence from the tumor matrix. In the marginal area, tumor cells emitted a strong fluorescence, but the cell count was less than that of tumor bulk. The tissue of the adjacent edematous brain demonstrated several small cells on H & E staining; however, there were no tumor cells detected as emitting fluorescence using the Cy 5 filter (Figure 3A–F).

### 3.3. Illustrative Cases

Case one: A 56 year-old woman had a cystic glioblastoma in the right frontal lobe, manifesting as a severe headache. The tumor was resected *en bloc* using an optical navigation system under continuous motor-evoked potential monitoring. The tumor was irradiated using a laser, and the resected tissue of the tumor cyst wall emitted strong red fluorescence upon observation using a surgical microscope, and fluorescence microscopy demonstrated that many tumor cells in the resected tissue also emitted strong fluorescence. After resection of the tumor bulk, the tissue surrounding the right frontal horn of the lateral ventricle emitted strong to weak fluorescence, and several tumor cells also emitted fluorescence, and were therefore was additionally resected. H&E-stained tissue demonstrated a few atypical cells in the edematous matrix. Postoperative contrast-enhanced MRI demonstrated complete resection of the tumor, and the additional resected area was clearly identified (Figure 4A–H).

Case ten: A 54 year-old man had a glioblastoma in the left temporal lobe, manifesting as a severe headache and a partial seizure. The tumor, which emitted strong fluorescence on surgical microscopy, was resected *en bloc* using an optical navigation system, and many atypically shaped tumor cells in the resected tissue also emitted strong fluorescence on the fluorescence microscopy. After resection of the tumor bulk, the tissue of the left hippocampus emitted very weak fluorescence, but several invading tumor cells emitting strong fluorescence were demonstrated on the fluorescence microscopy, and therefore the hippocampal head was partially resected. Postoperative contrast-enhanced MRI demonstrated complete resection of the tumor, and the additional resected area was clearly identified (Figure 5A–H).

Case eleven A 30 year-old man had an oligodendroglioma in the left frontal lobe, manifesting as a first episode of generalized tonic and clonic seizures. The tumor was resected *en bloc* under awake craniotomy, and emitted very weak fluorescence on surgical microscopy, with several tumor cells in the tumor bulk emitting strong fluorescence. However, the merged image of Cy 5 image and PDD-B fluorescence demonstrated a mixture of Cy 5-positive cells and Cy 5-negative cells (mosaic pattern) (Figure 6A–D).

Case eighteen: A 28 year-old man had a recurrent anaplastic oligodendroglioma in the left frontal lobe. The tumor was resected *en bloc*, which emitted extremely strong fluorescence on surgical microscopy, and many tumor cells in the tumor bulk emitted strong fluorescence. The merged image of Cy 5 and PDD-B fluorescence demonstrated that all tumor cells were Cy 5-positive (diffuse pattern) (Figure 6E–H).

### 3.4. Association between Fluorescence Intensity on Surgical Microscopy and Number of Cells Emitting Fluorescence from TPS

We resected the tumor bulk by photodiagnosis using surgical microscopy, and confirmed that the tumor cells emitted the strong fluorescence in the edematous brain matrix with the faint background fluorescence on fluorescence microscopy. There were more than 400 (median: 401; range: 75–1138) strong fluorescence-positive tumor cells in 100 microscopic fields, and large tumor cells showed granular aggregates in the cytoplasm emitting red fluorescence (Appendix A). We were able to confirm the localization of tumor cells by fluorescence emission from their cytoplasm, detected by TPS using the Cy 5 filter in combination with the fluorescence detected using the PDD-B filter. The wall of the resected tumor cavity usually shows weak fluorescence under surgical microscopy, and demonstrate the 50 to 200 (median: 88; range: 34–221) cells emitting the red fluorescence under the faint fluorescent background of the edematous brain matrix. The sizes of the cells emitting red fluorescence in this area were almost uniform. When we confirmed the disappearance of fluorescence in the marginal zone of resection on surgical microscopy, we confirmed the presence of fluorescence-positive tumor cells despite there being less than 50 (median: 26; range: 0–71) cells. Therefore, we performed resection of the peritumoral tissue as long as there were no problems with intraoperative physiological monitoring, but only four specimens resected from two patients in this study showed a complete disappearance of fluorescence-emitting tumor cells. We confirmed the statistically significant differences in fluorescence-emitting tumor cell counts among tissue resected from the regions emitting strong, weak and no fluorescence on surgical microscopy (Figure 7).

### 3.5. Improvement in the Extent of Resection of Malignant Gliomas Using This System

Evaluation of the extent of resection (EOR) by the postoperative MRI performed within 3 days after surgery demonstrated that the mean EOR of the 25 patients was 94.6 ± 5.6%, and the median EOR was 98%. Among the patients with newly diagnosed tumors, the mean EOR was 96.8% ± 3.5% and the median EOR was 100%, and in the patients with recurrent tumors, the mean EOR was 90% ± 7.3% and the median EOR was 89.5%. Regarding the patients for whom additional resection was performed according to the intraoperative fluorescence cytology findings, despite the disappearance of fluorescence under surgical microscopy, these were four newly diagnosed patients and one recurrent patient. In these patients, the postoperative MRI demonstrated “supra-total resection” of the tumor.

## 4. Discussion

In Wilson’s scheme of the degree and the distribution of the intracerebral invasion of glioblastoma cells, he suggested that the reduction of these invading tumor cells in the BTI is an important factor for improving the extent of resection of glioblastoma tissue [25]. Conventionally, the extent of resection of the tumor bulk evaluated the gadolinium-enhanced postoperative MRI was the most important prognostic factor [7,8,9,10]. The target of FGR using 5-ALA was the tumor bulk demonstrating gadolinium-enhancement on MRI, and this procedure was clinically approved as improving the progression-free survival of glioblastoma patients, but did not prolong the overall survival of these patients [7]. Therefore, in glioblastoma surgery, as much as possible of the surrounding tissues infiltrated by tumor cells should be resected, up to the boundary of the functional region of the brain [8,9,10,25]. Recently, the significance of aiming at the “supra-total” resection, which is resection of tissue containing the invading glioblastoma cells in the area demonstrating a high signal on the preoperative FLAIR imaging beyond the gadolinium-enhanced tumor bulk, was reported [5]. However, the subjective evaluation of fluorescence emission by the naked eye, the problems with the sensitivity and specificity of PPIX accumulation within tumor cells and the autofluorescence of PPIX from normal brain tissue were important clinical issues in performing the FGR using 5-ALA, particularly for additional resection of the region demonstrating “vague” fluorescence around the tumor bulk emitting strong fluorescence [8,9,10,11,12,13,14,15]. There have been numerous discussions regarding these unresolved problems to date.

Recently, several attempts to resolve these crucial issues of FGR to identify infiltrating the glioblastoma cells in BTI intraoperatively were carried out. For instance, the difference in the wavelength of the autofluorescence emitted from normal brain tissue and brain tumor tissue was analyzed [26,27,28], and emission “vague” fluorescence measured using the handheld-type spectrometer and confocal microscope in situ was analyzed [29,30,31,32,33,34,35,36,37,38,39]. However, the normal brain emits autofluorescence of three wavelengths, originating from: NAD(P)H (nicotinamide adenine dinucleotide) (ex. 360–380 nm, em. 450–500 nm); FAD (flavin adenine dinucleotide) (ex. 440–450 nm, em. 500–550 nm); and porphyrin (mainly PPIX) (ex. 490–500 nm, em. 590–630 nm) [40]. However, these wavelengths are strongly affected by the absorption of hemoglobin; therefore, it was not easy to detect tumor infiltration using autofluorescence analysis [26,27,28]. In addition, it was also considered that the evaluation using the handheld-type spectrometer and confocal microscopy was affected by the PPIX autofluorescence of the normal brain [29,30,31]. Furthermore, we have some concerns regarding the stability of intraoperative data acquisition and using these handheld-type compact devices. In this regard, our system appears to be simple and consistent for evaluating fluorescence emission of TPS from glioblastoma cells, because it has a wavelength range that is longer than the autofluorescence of the normal brain, which is called the “optical window”, an indication that it is minimally affected by the absorption of hemoglobin. We were in fact able to distinguish between various types of autofluorescence and the fluorescence emission of TPS from the cotton fiber and iodine egg shell by using the three-filter system in the phantom experiment. In addition, in the operating theater, we were able to clearly evaluate the autofluorescence of brain tissue and the fluorescence emission of TPS from glioblastoma cells on the PC monitor using the Cy5 filter system. Furthermore, the method of tissue preparation and the procedure of fluorescence microscopy are extremely simple, easy and stable, and the results can be confirmed within 2 min after a biopsy. In addition, all the staff, including the surgeon performing the surgery, can confirm the presence of tumor cells emitting red fluorescence, and can objectively determine the necessity of additional resection of the brain tissue. Furthermore, if a biopsy was performed on an area concluded as having no fluorescence by macroscopic photodiagnosis under surgical microscopy, and cells emitting the red fluorescence were still present under fluorescence microscopy [19,20], we performed additional resection if possible according to the patient’s neurophysiological evaluation, using PDT in functional brain areas.

Regarding the patients in this study, additional resection was performed on all patients according to the findings on fluorescence microscopy after macroscopic photodiagnosis, up to the limit of the resection evaluated by the intraoperative functional monitoring and navigation. MRI within 3 days after surgery demonstrated the area of additional resection, and we believe we achieved almost supra-total resection in all of our glioblastoma patients.

One patient with low-grade glioma demonstrated interesting findings regarding fluorescence-emitting cells. Almost all tumor cells of the tumor bulk of malignant gliomas demonstrated TPS uptake (diffuse pattern), but the tumor cells of the low-grade glioma were a mixture of the cells emitting TPS and those not emitting TPS (mosaic pattern). This finding indicates one of the basic mechanisms causing the weak or no fluorescence image on macroscopic photodiagnosis of low-grade glioma [19,20,39]. TPS, which is a water-soluble photosensitizer, promptly conjugates to albumin after its intravenous administration and funnels the blood vessel [41]. TPS is then distributed in the interstitial tissue after direct leakage from the disrupted blood–brain barrier (BBB) or leakage by bulk flow into the interstitial fluid, called the enhanced permeability and retention effect, and is taken up into tumor cells by the transporter SLC46A1, which is a heme carrier protein that accumulates in the lysosomes of tumor cells [42]. In this fluorescence microscopic study, strong red fluorescence was detected from the cytoplasm of the tumor cells, and faint fluorescence was also detected from the interstitial tissue of the tumor. The faint fluorescence might be emitted from the remnant TPS in the edematous interstitial tissue of the tumor which was not taken up by the tumor cells. In low-grade gliomas, faint fluorescence was also emitted from the interstitial tissue, but the tumor cells demonstrated a mosaic uptake pattern of TPS, which suggests the possibility of a heterogeneous uptake mechanism in tumor cells. Low-grade gliomas have minimal disruption of the BBB, fewer tumor cells and lower proliferation potency than high-grade gliomas, and are hence expected to demonstrate less fluorescence from the photosensitizer [19,20,42]. Although our findings suggest that the heterogeneous uptake mechanism of TPS plays an important role in the lower fluorescence emission from low-grade glioma tissue, in the future, further research should be performed to clarify the uptake mechanism of TPS in these tumor cells.

Regarding the limitations of this study, evaluations were limited to a small number of patients with glioma who underwent surgery in a single institution, many years ago. In the future, evaluations should be performed for all types of primary malignant brain tumors, including malignant meningiomas and malignant skull based tumors. Furthermore, data of surgeries performed by other surgeons in other institutions should be included from the viewpoint of objectivity, and to promote the spread of this brief, reliable and low-cost method for intraoperative brain tumor cell detection that we developed.

## 5. Conclusions

This intraoperative rapid fluorescence cytology system is expected to become useful in many areas of the cancer surgery in the future. As it is known that TPS is taken up and retained in the most cancer cells, this method will offer crucial additional information for surgeons, such as the confirmation of the accuracy of the biopsy sample, the limit of surgical resection and the presence of lymph node metastases within an extremely short time. Furthermore, all medical staff can objectively understand the importance of the surgery by sharing such information. The Japanese government approves the administration of TPS under medical insurance for surgeries for primary lung cancer and recurrent esophageal cancer after chemoradiotherapy, as well as primary malignant brain tumors. We believe that our brain tumor cell detection system will prove useful for all surgeons who treat these types of cancers.

## Figures and Tables

**Figure 1 jcm-10-05375-f001:**
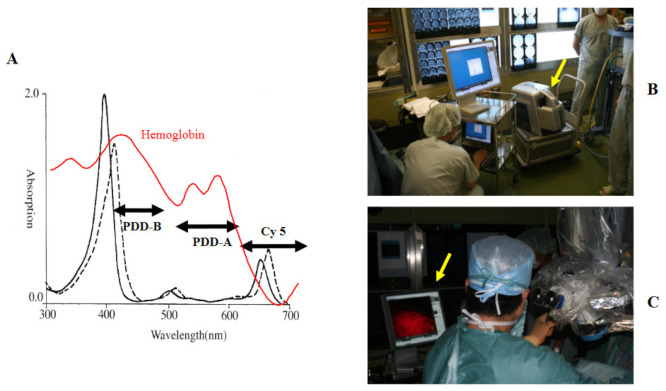
Absorption spectrum of TPS and filter systems. (**A**) Absorption spectrum of TPS and hemoglobin. (solid line: TPS and phosphate buffer solution; dotted line: TPS conjugated with albumin; red line: hemoglobin). Three filter systems were used to observe the fluorescence emitted from the obtained tissues. (PDD-B: excitation 450/40 nm, absorption 460 nm; PDD-A: excitation 550/40 nm, absorption 610 nm; Cy 5: excitation 620/60 nm, absorption 700/75 nm). (**B**) A compact fluorescence microscope (yellow arrow) placed in the operating theater. (**C**) Fluorescence images obtained under each filter system were displayed on the PC monitor (yellow arrow), and the surgeon was able to detect the presence of the tumor cells emitting fluorescence.

**Figure 2 jcm-10-05375-f002:**
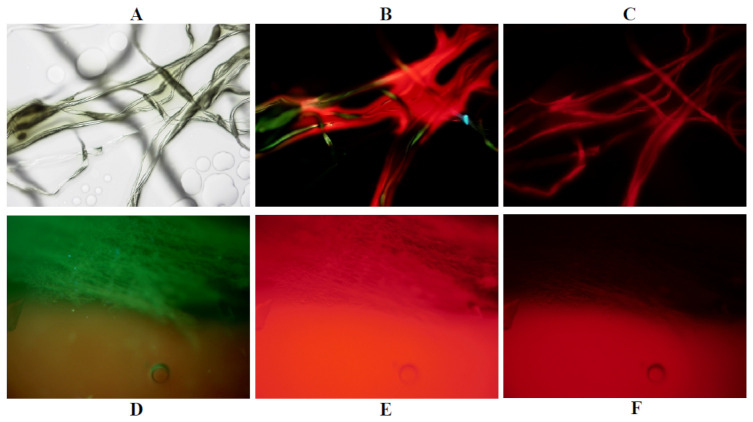
Phantom Experiments. (**A**–**C**) Cotton fibers were moistened with TPS and bovine serum. (**D**–**F**) Autofluorescence of PP IX from an iodine egg shell. (**A**) Image under white light; (**B**) Merged image under the PDD-B and PDD-A filters; (**C**,**F**) Under the Cy 5 filter; (**D**) Under the PDD-B filter; (**E**) Under the PDD-A filter.

**Figure 3 jcm-10-05375-f003:**
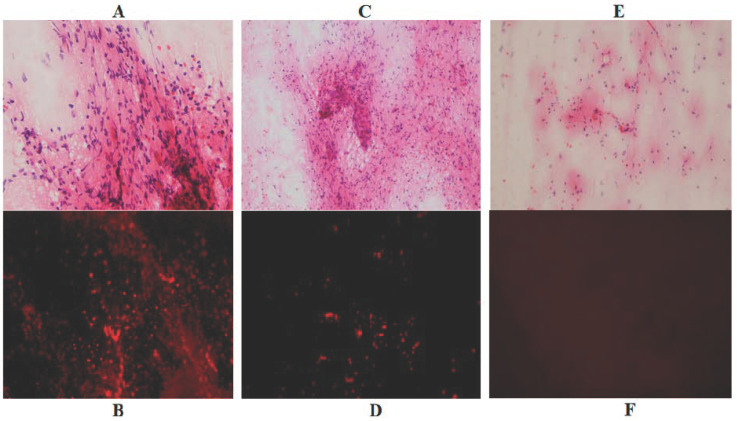
Intraoperative detection of fluorescence emitted from tumor tissues and adjacent brain tissue. (**A**–**C**) Hematoxylin and eosin staining of crush smear tissue samples of the tumor bulk (**A**); marginal area of the tumor (**B**); and adjacent edematous brain tissue (**C**). (**D**–**F**) Fluorescence images under the Cy 5 filter system. A total of 208 cells emitting strong fluorescence were detected in the tissue of tumor bulk (**D**); 114 cells were detected in the tissue of marginal area (**E**); and only a few cells were detected in the adjacent brain tissue (**F**).

**Figure 4 jcm-10-05375-f004:**
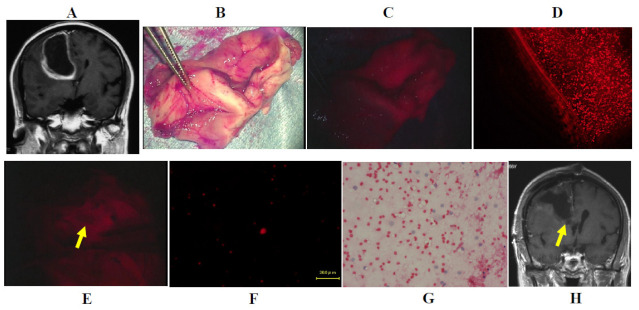
Illustrative case one. (**A**) Gadolinium-enhanced T1-weighted coronal image demonstrating a strongly enhanced cystic glioblastoma lesion in the right frontal lobe. (**B**) Brain tissue containing the tumor tissue was resected completely using an optical navigation system under continuous motor evoked potential monitoring. (**C**) Strong fluorescence from the resected tumor tissue was detected under a surgical microscope during photodiagnosis. (**D**) A total of 761 cells emitting fluorescence were detected in the tumor issue under the fluorescence microscopy using the Cy 5 filter. (**E**) Weak fluorescence (yellow arrow) was detected in the marginal brain tissue under surgical microscopy. (**F**) A total of 61 cells emitting fluorescence were detected in the marginal brain tissue under fluorescence microscopy using the Cy 5 filter. (**G**) Hematoxylin and eosin staining of a crush smear sample of the same tissue same as in (**F**). (**H**) Gadolinium-enhanced T1-weighted coronal image after surgery demonstrating complete resection of the tumor bulk and the additional resected area (yellow arrow) of the marginal brain tissue.

**Figure 5 jcm-10-05375-f005:**
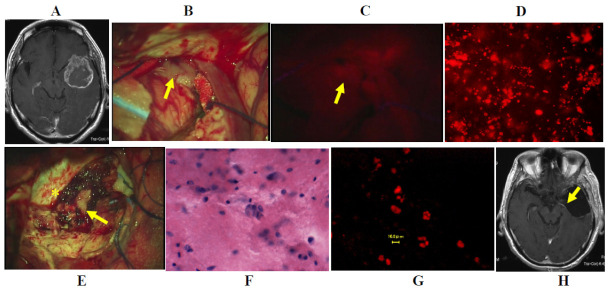
Illustrative case ten. (**A**) Gadolinium-enhanced T1-weighted axial MR image demonstrating a heterogeneously enhanced glioblastoma lesion in the left temporal lobe. (**B**) The tumor tissue showed exophytic growth in the left temporal lobe (yellow arrow) on the surgical microscopy. (**C**) Strong fluorescence was detected on surgical microscopy (yellow arrow) for photodiagnosis. (**D**) A total of 412 polymorphic tumor cells emitting fluorescence were detected in the tumor bulk tissue under fluorescence microscopy using the Cy 5 filter. (**E**) After resection of the tumor bulk, the hippocampal head (yellow arrow) showed a normal appearance under surgical microscopy. (**F**). Hematoxylin and eosin staining of crush smear tissue of the hippocampal head demonstrated infiltrating multinucleated tumor cells under the white light of the fluorescence microscopy. (**G**). Infiltrating tumor cells in the tissue in (**F**) demonstrated strong fluorescence on the fluorescence microscopy using the Cy 5 filter. (**H**). Postoperative gadolinium-enhanced T1-weighted axial MR image demonstrated complete resection of the tumor bulk and additional resection of the hippocampal head (yellow arrow).

**Figure 6 jcm-10-05375-f006:**
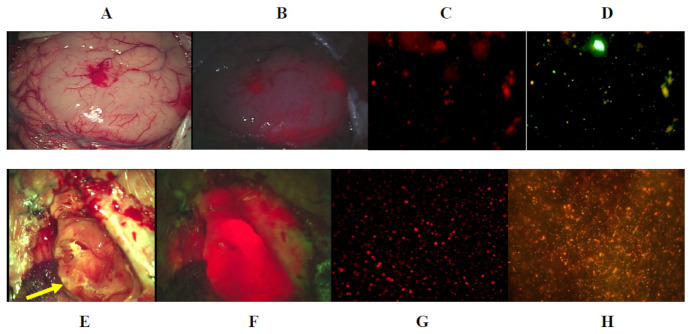
Illustrative case eleven (**A**–**D**) and case eighteen (**E**–**H**). (**A**) The tumor tissue in the cortical gyrus of the left frontal lobe showed hypertrophic features. (**B**) Weak fluorescence was detected on photodiagnosis using the surgical microscopy. (**C**) A total of 97 cells emitting fluorescence were detected in the tumor bulk tissue under fluorescence microscopy using the Cy 5 filter. (**D**) Merged images taken using the Cy 5 filter and the PDD-B filter demonstrated a mosaic pattern of fluorescence-emitting and non-emitting cells. (**E**) The recurrent tumor tissue was a nodular cortical mass (yellow arrow). (**F**) Strong fluorescence was observed on photodiagnosis using surgical microscopy. (**G**) A total of 573 cells emitting fluorescence were detected in the tumor bulk under fluorescence microscopy using the Cy 5 filter. (**H**) Merged image of images taken using the Cy 5 filter and PDD-B filter demonstrated a diffuse pattern, with almost all tumor cells emitting fluorescence.

**Figure 7 jcm-10-05375-f007:**
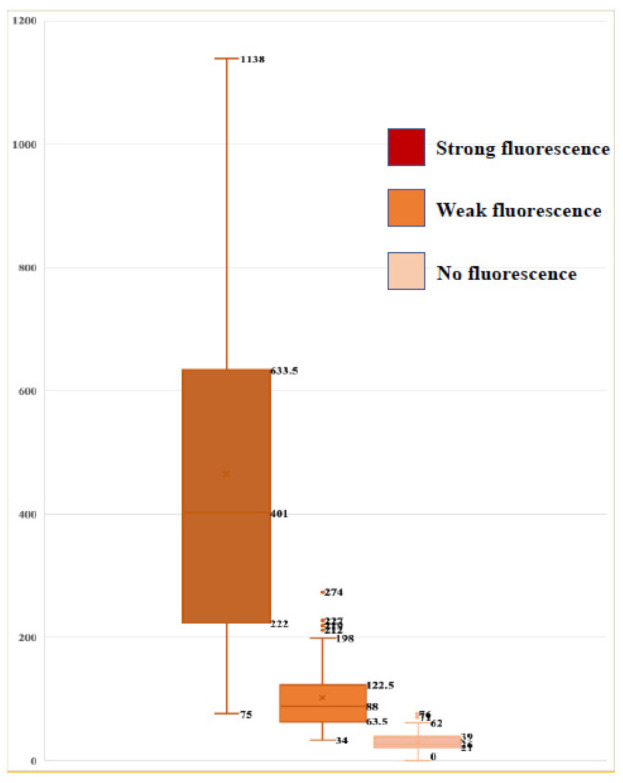
Association between fluorescence intensity on photodiagnosis using surgical microscope, and cell count determined by fluorescence emission on fluorescence microscopy. There was a statistically significant difference between the strong fluorescence and the weak fluorescence group (*p* < 0.001), and the weak fluorescence and no fluorescence group (*p* < 0.05) (Student *t*-test).

## Data Availability

The study did not report any data.

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
