# Peer review of "Preliminary Report: Rapid Intraoperative Detection of Residual Glioma Cell in Resection Cavity Walls Using a Compact Fluorescence Microscope"

_jcm, 2021, doi:10.3390/jcm10225375_

Round 1

Reviewer 1 Report

An interesting and fascinating article to read. The methods section is well explained, some doubts and issues:

It is wrong to claim that fluorescence-guided resection is recently proposed. In fact, since the 80's there are works that talk about this resection with fluorescence and later with the use of 5-ALA.

It is necessary to better describe in which phases of the surgical excision procedure the laser is used and if its usefulness can be defined as a "final check of the margins" and if according to the authors it can have a support for a supramarginal resection.

For periventricular lesions, what type of signal is received? For very bleeding lesions?

Need a more detailed report of any reported side effects.

Author Response

Reviewer #1 (Reviewer Comments to the Author)

An interesting and fascinating article to read. The methods section is well explained, some doubts and issues:

We really appreciated the review for his careful evaluation of our manuscript.

Comment 1

It is wrong to claim that fluorescence-guided resection is recently proposed. In fact, since the 80’s there are works that talk about this resection with fluorescence and later with the use of 5-ALA.

Reply to comment 1

We wish to thank the reviewer for this comment. We never denied this method and bring up some problems of FGR using 5-ALA and by overcoming them, expect further contribution in the brain tumor resection. In accordance with the reviewer’s comment, we have changed Line 81-85. “Therefore, basic and clinical studies in quantification of the 5-ALA metabolite protoporphyrin IX (PPIX) in the brain and tumor tissues using optical spectroscopy have been performed, and intraoperative augmentation methods of this information using artificial intelligence have been developed to overcome these problem.” to “ To overcome these problems, several authors reported the basic and clinical studies on quantification of the 5-ALA metabolite protoporphyrin IX (PPIX) in the brain and tumor tissues using optical spectroscopy, and intraoperative augmentation method of these optical information using artificial intelligence.”

Comment 2

It is necessary to better describe in which phases of the surgical excision procedure the laser is used and if its usefulness can be defined as a “final check of the margins” and if according to the authors it can have a support for a supramarginal resection.

Reply to comment 2

Thank you for your important comment. As described in the part of method, intraoperative photodiagnosis using a surgical microscope (Line 119 to 130), photodiagnosis using laser was performed appropriately after brain is exposed until we finish resecting the tumor. The details of this procedure was published in our own paper, reference number 19, entitled “ Intraoperative photodiagnosis for malignant glioma using photosensitizer talaporfin sodium”.

We appreciate the reviewer’s comment that our method was useful to detect the residual tumor cells as final check of the resection cavity wall. And we described in the part of discussion (Line 364-369) that additional resection was performed on all patients of this study according to the findings of our method and we believe that we achieved almost supra-total resection demonstrated on MRI within 3 days after surgery.

Comment 3

For periventricular lesions, what type of signal is received? For very bleeding lesions?

Reply to comment 3

We thank the reviewer for this comment. When we reached the paraventricular lesion, the intensity of fluorescence detected on surgical microscope depend on the density of tumor cells. We usually identify the weak fluorescence which seemed to be emitted from the tissue of edematous matrix with a few tumor cells. Please find the Fig.3 B, the weak fluorescence was demonstrated from the edematous brain matrix. And Fig.5 E and F demonstrated the weak fluorescence from the tissue of surrounding the right frontal horn of the lateral ventricle containing a few tumor cells. And these tissues were not usually easy to bleed.

Comment 4

Need a more detailed report of any report of side effects.

Thank you for your comment. It is performed in approximately same protocol of PDD and PDT using talaporfin sodium in several reports. The major adverse event was not reported and the minor adverse event such as skin photosensitivity and mild liver dysfunction was reported so far. In this article, the dose of talaporfin sodium was equivalent to these report and the purpose of the article was pathological diagnosis using fluorescence microscopy, therefore, we did not mention side effect.

Reviewer 2 Report

The authors determined the correlation between tumor cell density by fluorescent microscope and smear preparation of tumor tissue of 25 patients with malignant glioma. The authors found a high association  with the strength of fluorescence and the amount of remaining tumor cells and conclude this method to provide crucial information for surgical resection. The authors are to be applauded for their detailed descriptions of the novel approach and their solid experimental approach. 

Would recommend emphasizing in the Discussion section how this novel fluorescent approach is different than the existing ones in the literature (e.g. 5-ALA).

Author Response

Reviewer #2 (Reviewer Comments to the Author)

The authors determined the correlation between tumor cell density by fluorescent microscope and smear preparation of tumor tissue of 25 patients with malignant glioma. The authors found a high association with the strength of fluorescence and the amount of remaining tumor cells and conclude this method to provide crucial information for surgical resection. The authors are to be applauded for their detailed descriptions of the novel approach and their solid experimental approach.

We really appreciate the reviewer for his careful evaluation of our manuscript.

Comment 1

Would recommend emphasizing in the discussion section how this novel fluorescent approach is different than the existing ones in the literature (e.g. 5-ALA).

Thank you for the generous evaluation. Our method is very simple. We just evaluated the photosensitizer which specifically accumulate to tumor cells by a fluorescence microscope. Especially, our method is using a very compact fluorescence microscope in the operating theater, and it can evaluate the fluorescence emitting from the tumor tissues rapidly, and a surgeon can evaluate it with own eyes. Anyone who is in the operating room can share theevaluation without high costing. We are thinking with a groundbreaking system in the

accomplishment of the fluorescence diagnosis of the brain tumor.

Reviewer 3 Report

This topic about talaporfin sodium and its application in 25 patients appears to be interesting, please look at these points:

  1. Line 152: I'm unable to see table 1 in the manuscript, did authors forget to report it in the manuscript?
  2. Lines 32-33: "In the management of the malignant gliomas, the extent of the surgical resection is the most important prognostic factor [1-5]". This sentence is correct  
  3. Lines 337-339: "Recently, the significance of aiming at the “supra-total” resection... FLAIR imaging beyond the gadolinium-enhanced tumor bulk was reported" As your results reported a good supra-total resection, I think this part should be moved in the introduction section.
  4. Can this technique also be used during a stereotactic biopsy to assess whether the fragment is pathological?
  5. Lines 214-273. "3.3. Illustrative cases" section. Please, order cases in the following way: case 1, case 10, case 11, case 18.
  6. Lines 347-380: In the discussion section, please, report (just few lines) the role of glioblastoma microenvironment and fluorophores.
  7. Line 412: "single institution and too many years ago". What do authors want to say about "too many years ago"? Is this a limitation in your opinion? 
  8. Line 411: "glioma who underwent surgery performed by a single surgeon (JA)" . I don't think this can represent a limitation of the study, but rather a strength, as it indicates a standardization of the procedure.

Overall a good paper.

Author Response

Reviewer #3 (Reviewer Comments to the Author)

This topic about talaporfin sodium and it application in 25 patients appears to be interesting. Please look at these points.

Comment 1

Line 152: I’m unable to see table 1 in the manuscript, did authors forget it in the manuscript?

Reply to comment 1

Thank you for your comment. Table 1 is supplementary material. In accordance of reviewer’s comment, we have changed table 1 to supplementary table 1.

Comment 2

Line 22-34: In the management of the malignant gliomas, the extent of the surgical resection in the most important prognostic factor [1-5]. This sentence is correct.

Reply to comment 2

We wish to thank the reviewer for this comment.

Comment 3

Line 337-339: Recently, the significance of aiming at the supra-total resection.

FLAIR imaging beyond the gadolinium-enhanced tumor bulk was reported. As your results reported a god supra-total resection, I think this part should be moved in the introduction section.

Reply to comment 3

Thank you for your comment. In the patients in this study, additional resection was performed on all patients according to the findings on fluorescence microscopy. MRI within 3 days after surgery demonstrated the area of additional resection, and we believe we achieved supra-total resection using this method. This finding is a most important point of our article, therefore we describe this in the result and discussion part.

Comment 4

Can this technique also used during a stereotactic biopsy to assess whether the fragment is pathological?

Reply to comment 4

Thank you for your valuable comment. This method will offer crucial additional information such as the confirmation of the accuracy of the biopsy sample, the limit of surgical resection, and the presence of lymph node metastases within a extremely short time.

Comment 5

Line 214-273. 3.3 illustrative cases section. Please order cases in the following way: Case 1, Case 10, Case 11, Case 18

Reply to comment 5

Thank you for your comment. According to reviewer’s opinion, we revised the numbering of cases on supplementary table 1.

Case 6

Line 347-380: In the discussion section, please, report (just few lines) the role of glioblastoma microenvironment and fluorophores.

Reply to comment 6

Thank you for your crucial suggestion. In the tissue of brain-tumor interface, the strong red fluorescence was detected from the cytoplasm of the tumor cells, and faint fluorescence was detected from the interstitial tissue of the tumor. (Fig.3 B and D) The faint fluorescence was emitted from the remnant talaporfin sodium in the edematous interstitial tissue.

Comment 7

Single institution and too many years ago. What the authors want to say about “ too many years ago?” Is this limitation in your opinion?

Reply to comment 7

Thank you for your comment. The case group that we used in this study was actually the case group that they operated for more than 10-15 years ago.  We did not conduct a recent case and, in various circumstances, did whether even a recent case turned out similar with limitation because it was unknown.

Comment 8

Line 412 ; gloma who underwent surgery performed by a single surgeon (JA), I don’t think this can represent a limitation of the study, but rather a strength, as it indicates a standardization of the procedure.

Reply to comment 8

Thank you for your comment. In accordance of reviewer’s comment, we delete this sentence.

Round 2

Reviewer 1 Report

Acceptable

Reviewer 3 Report

Authors solved all my criticisms.

This manuscript is a resubmission of an earlier submission. The following is a list of the peer review reports and author responses from that submission.

Round 1

Reviewer 1 Report

  In this single-center study, the authors present the case of 25 consecutive patients who underwent craniotomy for malignant glioma using a novel technological strategy to maximize the extent of resection. This strategy consists of analyzing the already described fluorescence-guided resection (FGR) method with a new photosensitizer (in replacement of 5-ALA), talaporfin sodium (TPS), coupled to a semiconductor laser system. The authors conclude that this novel technique is rapid to detect the presence of tumor cells within the cavity walls, and that it provides opportunity for a greater extent of resection (EOR).

This is an interesting manuscript that presents a novel strategy to evaluate the presence of tumor cells within the resection cavity walls and might be useful in the optimization of the EOR. Some issues should be addressed to make this manuscript more suitable for publication:

  • The authors state that patients underwent resection under “physiological monitoring”, and that they “performed resection of the peritumoral tissue as long as there was [sic] no problems on intraoperative physiological monitoring”. It would be important to precise the nature of this monitoring for all included patients.
  • Although the EOR results are encouraging, patients are not compared with an appropriate control group. This would be important to be able to have a better evaluation of this novel resection strategy.
  • Although the EOR results are encouraging, this alone cannot be interpreted. The EOR must be compared with the amount of normal parenchyma resected and clinical implications of this resection to be relevant. This data is not presented in the manuscript.
  • It would be relevant to show data on the clinical outcomes of these patients, for example with the progression-free survival and a functional scale to illustrate the clinical impact of this novel resection strategy.

Author Response

Reviewer #1 (Reviewer Comments to the Author):

  In this single-center study, the authors present the case of 25 consecutive patients who underwent craniotomy for malignant glioma using a novel technological strategy to maximize the extent of resection. This strategy consists of analyzing the already described fluorescence-guided resection (FGR) method with a new photosensitizer (in replacement of 5-ALA), talaporfin sodium (TPS), coupled to a semiconductor laser system. The authors conclude that this novel technique is rapid to detect the presence of tumor cells within the cavity walls, and that it provides opportunity for a greater extent of resection (EOR).

  This is an interesting manuscript that presents a novel strategy to evaluate the presence of tumor cells within the resection cavity walls and might be useful in the optimization of the EOR. Some issues should be addressed to make this manuscript more suitable for publication:

  We really appreciate the reviewer for his careful evaluation of our manuscript.

Comment 1

  The authors state that patients underwent resection under “physiological monitoring“, and that the “performed resection of the peritumoral tissue as long as there was no problems on intraoperative physiological monitoring“. It would be important to precise the nature of this monitoring for all included patients.

Thank you for the valuable comment. The extent of resection (EOR) of glioblastoma were depend on the intraoperative functional evaluation by motor evoked potential (MEP) monitoring or estimation of the speech function by awake surgery. Therefore, we conducted the MEP for many cases in this study except for the cases of tumors locating the occipital lobe and cerebellum, and performed awake surgery in the cases of the tumor locating the left front-temporal lobes. The main purpose of our study is emphasize the clinical feasibility of the intraoperative estimation of the residual glioma cells in the resection cavity using a compact fluorescence microscope after performing the resection of the tumor to the functional limit according to the estimation of these physiological monitoring. Therefore, we have not mentioned about the methods of these functional estimation for each cases.  

Comment 2

  Although the EOR results are encouraging, patients are not compared with an appropriate control group. This would be important to be able to have a better evaluation of this novel resection strategy.

Thank you for the valuable indication. We evaluated the extent of resection (EOR) of the cases of malignant glioma by postoperative MRI within 3 days after surgery using intraoperative fluorescence detection from the residual cells in resection cavity walls using a compact fluorescence microscope, and EOR of out series was very high. As reviewer mentioned, it is necessary to compare the result of EOR of the cases of malignant gliomas resecting without using this method, for emphasize the utility of this method.  However, we suggest the readers who can read our article knowing the historic data about the EOR of glioblastoma. In historical data, the rate of the cases that is available for complete removal in around 50% among the glioblastoma cases, and it is only 56.4% that obtained 95% or more of resection in 500 cases analysis from UCSF which is the highest volume center of the glioma surgery in the world. (Reference 2) Of course, the cases of our study intended for selected cases predicting the availability for complete removal from the preoperative radiological findings. We would like to emphasize the utility of the developed intraoperative detection methods of the fluorescence emitting from the residual glioma cells in the resected cavity walls for those cases, obtaining a high EOR surely.

Comment 3

  Although the EOR results are encouraging, this alone cannot be interpreted. The EOR must be compared with the amount of normal parenchyma resected and clinical implications of this resection to be relevant. This data is not presented in the manuscript.

Thanks for valuable comment. We previously described in the answer for Comment 1, but we decided the limit of resection of the tumor according to the intraoperative brain functional evaluation using MEP or awake surgery. And fortunately, there was not the case that showed the postoperative neurological complications. However, we do not mention about these clinical results in this manuscript. Because our purpose of this study was demonstrating the clinical utility of the developed intraoperative method for detecting the residual glioma cells using fluorescence microscope for improvement the EOR of glioblastoma.

Comment 4

  It would be relevant to show data on the clinical outcomes of these patients, for example with the progression-free survival and a functional scale to illustrate the clinical impact of this novel resection strategy.

Thank you for pointing it out. The purpose of this article, as we mentioned previously, was whether the developed method contributes to improvement in EOR of the glioblastoma. We think that the improvement of the EOR of glioblastoma indicate the improvement of PFS and OS according to historical report by all means. However, the twenty-five cases in this article were not treated with the same protocol, such as the cases that treated PDT or not, or the cases using Temozolomide or not. Therefore, we could not show the clinical data such as PFS and OS of these patients. However, the cases received PDT including our cases, we have already reported an article. (Reference 21)

Reviewer 2 Report

The paper is well written and explained, introducing an innovative method at relatively low cost. As an initial study, it seems to me to be well conducted. I would dwell a bit more on the clinical aspects of selection and emphasize the issue of peri-ventricular lesions and in gliomas of eloquent area.

Author Response

Reviewer #2 (Reviewer Comments to the Author):

  The paper is well written and explained, introducing an innovative method at relatively low cost. As initial study, it seems to me to be well conducted.

We really appreciate the reviewer for his careful evaluation of our manuscript.

Comment 1

I would dwell a bit more on the clinical aspect of selection and emphasize the issue of peri-ventricular lesions and in gliomas of eloquent area.

Thank you for pointing it out. The twenty-five cases in this manuscript were selected the case that it is expected to complete resection by the preoperative radiological findings. However, as you know, the preoperative diagnosed patients with glioblastoma, almost all of the tumors reach the peri-ventricular lesion with diameters 4-5cm or more. About these cases, we evaluate the possibility of gross total resection of the tumor taking into consideration of the relationship with the functional eloquent area and the invasive degrees estimating from the preoperative radiological findings. Therefore, in abstract and main manuscript, we described it with twenty-five consecutive patients, but want to change to selectedpatients.

Reviewer 3 Report

the authors resent the experience of 25 patients glioma patients undergoing resection with the help of a fluorescence microscope

it is a bit irritating that the authors report on a consecutive series over a time span of 5 years, i.e. they operated only 5 patients per year, so there is the question about consistency

additionally irritating is that the authors report that the operations were done more than 11 to 16 years before manuscript submission (2006-2010) - I would highly recommend to add further patients being operated more recently

is there a clear definition of surgical strategy in regard to enlarging the extent of resection due to fluorescence, i.e. is the resection initially performed without fluorescence up to the stage when the surgeon thought he had achieved the maximum safe resection and then additionally fluorescnce analysis resulted in further resection - this would be the only way to determine the actual value of fluorescence

what was the neurological outcome of the patients (also in regard to enlarged resections - supra-marginal resection?)

 what is the WHO grade in the patients with enlarged resections?

is there a correlation of fluorescenec and WHO grading?

maybe it would make sense to concentrate the paper on a larger series of high-grade tumor operated with this technique

the authors should add survival data on the patients

the authors mention a study approval number of T2020 - does this reflect that the approval is from the year 2020? 

Author Response

Reviewer #3 (Reviewer Comments to the Author):

  The authors present the experience of 25 glioma patients undergoing resection with the help of a fluorescence microscope.

We really appreciate the reviewer for his careful evaluation of our manuscript.

Comment 1

  It is a bit irritating that the authors report on a consecutive series over a time span of 5 years, i.e., they operatied only 5 patients per year, so there is the question about consistency.

We would like to express my gratitude for valuable indication. We had described it in abstract and main manuscript with 25 consecutive patients, but there were selected patients actually predicted that the tumor will be perform the complete removal from the preoperative radiological findings, and decided to administrate the talaporfin sodium. Therefore, we would like to change from the consecutive patients to selected patients.

Comment 2

  Additional irritating is that the authors report that the operations were done more than 11 to 16 years before manuscript submission (2006-2010)- I would highly recomment to add further patients being operated more recently.

Thank you for the severe comment. As reviewer mentioned, ten years or more passed from the time when we performed this study.  During this period, we also developed the methods of photodiagnosis and photodynamic therapy for malignant glioma using talaporfin sodium, and moved into action to get health insurance approvement in our country. And we made certain experimental and clinical results, and these methods were able to perform the covering with health insurance approvement from Japanese Government. This is the reason for the delay for submit this article. It is a very shameful thing as authors, but please accept it.

Comment 3

 Is there a clear definition of surgical strategy in regard to enlarging the extent of resection due to fluorescence, i.e. is the resection initially performed without fluorescence up to the stage when the surgeon thought he had achieved the maximum safe resection and then additionally fluorescence analysis resulted in further resection- this would be the only way to determine the actual value of fluorescence.

Thanks for your valuable comment. Exactly. At first, we perform photodiagnosis under the operating microscope, and we resected the tumor bulk. Subsequently, the fluorescence from the tumor tissue becomes the weak, vague fluorescence under the operating microscope, and it is difficult to judge whether the tissue should be resect of not. At this stage, we emphasized the utility of the developed intraoperative detection the emitting fluorescence from residual tumor cells in the wall of resected cavity using a compact fluorescence microscope. In other words, at first, we perform photodiagnosis under surgical microscope which is macroscopic and are a flow to perform photodiagnosis under fluorescence microscope which is microscopic subsequently. We performed microscopic photo-diagnosis to the part of tumor bulk with strong fluorescence under macroscopic photo-dianosis, and could make the convincing evidence of being able to evaluate talaporfin sodium taken in the glioma cells.  As a result, we can demonstrate that it correlates with the cell count of the tumor cells which the strength of fluorescence that we evaluated in macroscopic photodiagnosis which is present in Fig. 4.

Comment 4

  What was the neurological outcome of the patients (also in regard to enlarged resections-supra-marginal resection?)

Thanks for your comment. We decided the limit of resection of the tumor according to the intraoperative brain functional evaluation using MEP or awake surgery. And fortunately, there was not the case that showed the postoperative neurological complications. However, we do not mention about these clinical results in this manuscript. Because our purpose of this study was demonstrating the clinical utility of the developed intraoperative method for detecting the residual glioma cells using fluorescence microscope for improvement the EOR of glioblastoma.

Comment 5

  What is the WHO grade in the patients with enlarged resections?

Thanks for your comment. Among twenty-five cases in this study, only one case is Grade II, and grade III is seven and grade IV is seventeen cases. It is one case of grade III and three cases of grade IV that were able to take it as supra-total resection by a postoperative MRI evaluation.

Comment 6

  Maybe it would make sense to concentrate the paper on a larger series of high-grade tumor operated with this technique.

Thank you for your comment. As reviewer mentioned, to show the utility of the developed our method, indicating the consistent results in more cases is necessary. We describe it as a limitation in our study in discussion part, indicating the results in more cases and in more institutions is necessary enough.

Comment 7

  The authors should add survival data on the patients.

Thanks for your important comment. The purpose of this article, as we mentioned previously, was whether the developed method contributes to improvement in EOR of the glioblastoma. We think that the improvement of the EOR of glioblastoma indicate the improvement of PFS and OS according to historical report by all means. However, the twenty-five cases in this article were not treated with the same protocol, such as the cases that treated PDT or not, or the cases using Temozolomide or not. Therefore, we could not show the clinical data such as PFS and OS of these patients. However, the cases received PDT including our cases, we have already reported an article. (Reference 21)

Comment 8

  The authors mention a study proposal number T2020- does this reflect that the approval is from the year 2020?

Thank you for important comment. As for this thesis, IRB in Tokyo Medical University was approved with No. 419 at first. However, it is necessary to obtain new approval in the IRB by the new regulations on clinical research when the revision of the clinical organon is in 2019 years in Japan and reports an article after 2020. Thus, it was necessary to obtain IRB approval again for submit this article. We would appreciate it if you can understand this situation.

Round 2

Reviewer 1 Report

Revisions acceptable

Reviewer 3 Report

-